# Direct modulation of GFAP-expressing glia in the arcuate nucleus bi-directionally regulates feeding

Naiyan Chen[1,2], Hiroki Sugihara[3†], Jinah Kim[2†], Zhanyan Fu[2,4†], Boaz Barak[2], Mriganka Sur[3], Guoping Feng[2,4*‡], Weiping Han[1*‡]

[1]Laboratory of Metabolic Medicine, Singapore Bioimaging Consortium, A*STAR, Singapore, Singapore; [2]Department of Brain and Cognitive Sciences, McGovern Institute for Brain Research, Massachusetts Institute of Technology, Cambridge, United States; [3]Department of Brain and Cognitive Sciences, Picower Institute for Learning and Memory, Massachusetts Institute of Technology, Cambridge, United States; [4]Stanley Center for Psychiatric Research, Broad Institute of MIT and Harvard, Cambridge, United States

*For correspondence: fengg@mit.edu (GF); weiping_han@sbic.a-star.edu.sg (WH)

[†]These authors contributed equally to this work
[‡]These authors also contributed equally to this work

Competing interests: The authors declare that no competing interests exist.

**Abstract** Multiple hypothalamic neuronal populations that regulate energy balance have been identified. Although hypothalamic glia exist in abundance and form intimate structural connections with neurons, their roles in energy homeostasis are less known. Here we show that selective $Ca^{2+}$ activation of glia in the mouse arcuate nucleus (ARC) reversibly induces increased food intake while disruption of $Ca^{2+}$ signaling pathway in ARC glia reduces food intake. The specific activation of ARC glia enhances the activity of agouti-related protein/neuropeptide Y (AgRP/NPY)-expressing neurons but induces no net response in pro-opiomelanocortin (POMC)-expressing neurons. ARC glial activation non-specifically depolarizes both AgRP/NPY and POMC neurons but a strong inhibitory input to POMC neurons balances the excitation. When AgRP/NPY neurons are inactivated, ARC glial activation fails to evoke any significant changes in food intake. Collectively, these results reveal an important role of ARC glia in the regulation of energy homeostasis through its interaction with distinct neuronal subtype-specific pathways.

## Introduction

The central nervous system comprises an elaborate network of neuronal populations that have been identified to control energy homeostasis (*Morton et al., 2006*; *Pang and Han, 2012*). This extensive network includes neural circuits that operate within (*Aponte et al., 2011*; *Atasoy et al., 2012*; *Jennings et al., 2013*; *Klöckener et al., 2011*; *Krashes et al., 2011*; *Lee et al., 2013*; *Vong et al., 2011*; *Yamanaka et al., 2003*) and beyond (*Georgescu et al., 2005*; *Hommel et al., 2006*; *Zhan et al., 2013*) the hypothalamus. Of these, the arcuate nucleus of the hypothalamus (ARC) is most actively studied. The key components of the feeding circuit in ARC have been identified to comprise agouti-related protein/neuropeptide Y (AgRP/NPY)-expressing neurons and pro-opiomela-nocortin (POMC)-expressing neurons (*Aponte et al., 2011*; *Atasoy et al., 2012*; *Krashes et al., 2011*; *Zhan et al., 2013*), although an earlier study (*Kong et al., 2012*) suggests other cell types may function as additional regulators. In particular, little is known about the functional role of glial fibrillary acidic protein (GFAP)-expressing glia that co-exist in the same brain circuit as these neurons.

Glia represent a major cell population in the brain, with their numbers at least equaling those of neurons. Traditionally viewed as passive support elements, recent research has unraveled their

**eLife digest** Neurons in an area of the brain called the hypothalamus control how much an animal eats. However, it is not clear what role other brain cells, such as glial cells, might play in influencing feeding. Glial cells do not send nerve impulses like neurons, but instead they mostly serve to support and protect the neurons.

Now, Chen et al. changed the activity of a particular kind of glial cell, known as astrocytes, to explore what effect this has on how much mice eat. Astrocytes are unique amongst glial cells because they can respond to neuronal activity and release chemicals that change the activity of other cells, including neurons.

The experiments revealed that switching astrocytes on in the hypothalamus made mice eat more, while turning them off had the opposite effect and reduced feeding. Chen et al. also found that glial cells partner with and change the activity of a particular group of neurons, known as the AgRP/NPY-expressing neurons. These neurons were already known to increase feeding activity when they become more active.

In contrast, Chen et al. showed that glial cells do not affect the activity of another group of neurons, known as POMC-expressing neurons. Previous research had shown that mice eat less when their POMC-neurons are more active.

Together the findings reveal that, within the hypothalamus, an interaction between glial cells and neurons influences how much an animal will eat. Further work is now required to understand the exact interaction between the glial cells and neurons, and to find out if other kinds of glial cells also have a role in controlling feeding.

important roles in the modulation of multiple physiological functions (*Chen et al., 2012*; *Gourine et al., 2010*; *Halassa et al., 2009*; *Nedergaard et al., 2003*; *Schummers et al., 2008*). GFAP-expressing hypothalamic glia, particularly astrocytes, have extensive structural connections with neurons (*Horvath et al., 2010*) and also express the receptor for the satiety hormone, leptin (*Cheunsuang and Morris, 2005*; *Hsuchou et al., 2010*, *2009*; *Jayaram et al., 2013*; *Kim et al., 2014*; *Pan et al., 2008*) and for insulin (*García-Cáceres et al., 2016*). Another type of GFAP-expressing hypothalamic glia, tanycytes, are interestingly glucosensitive and are activated by transmitters related to arousal and feeding (*Bolborea and Dale, 2013*; *Dale, 2011*; *Rodríguez et al., 2005*). These findings suggest a potential role of glia in the regulation of energy homeostasis, thereby opening up several key questions: Do GFAP-expressing hypothalamic glia play a role in the regulation of feeding? Do they functionally interact with AgRP/NPY and/or POMC neurons? If so, how does manipulation of glia in ARC affect food intake and feeding behavior? Here we demonstrate that direct and acute glial activation in the mouse ARC facilitates the activity of AgRP/NPY but not POMC neurons and is sufficient to reversibly evoke feeding. In the presence of AgRP/NPY neuronal inactivation, acute ARC glial activation did not evoke any change in feeding. In addition, disruption of $Ca^{2+}$ signaling in ARC glia leads to reduced food intake. This reveals an important role of ARC glia-AgRP/NPY neuron circuit in the regulation of energy homeostasis.

## Results

### Selective expression and activation of DREADDs in ARC glia

To acutely activate glia *in vivo*, we selectively expressed designer receptors exclusively activated by designer drugs (DREADDs) (*Alexander et al., 2009*; *Armbruster et al., 2007*) in GFAP-expressing glia in ARC of C57BL/6 adult mice. This was achieved by stereotaxic injection in ARC with an adeno-associated virus (AAV) containing a *Gfap* promoter-driven gene that encodes the evolved human M3-muscarinic receptor fused to mCherry (hM3D(Gq)-mCherry) (*Figure 1A*). The hM3D(Gq) couples to the Gq protein-mediated signaling to activate glia upon binding of clozapine-N-oxide (CNO). CNO is an exclusively specific ligand of DREADD (*Agulhon et al., 2013*) and is inert when used at low concentrations (see CNO dose experiments in *Figure 2—figure supplement 1B–D*). Successful bilateral viral injection in ARC was confirmed by localized expression of mCherry (*Figure 1B*). The

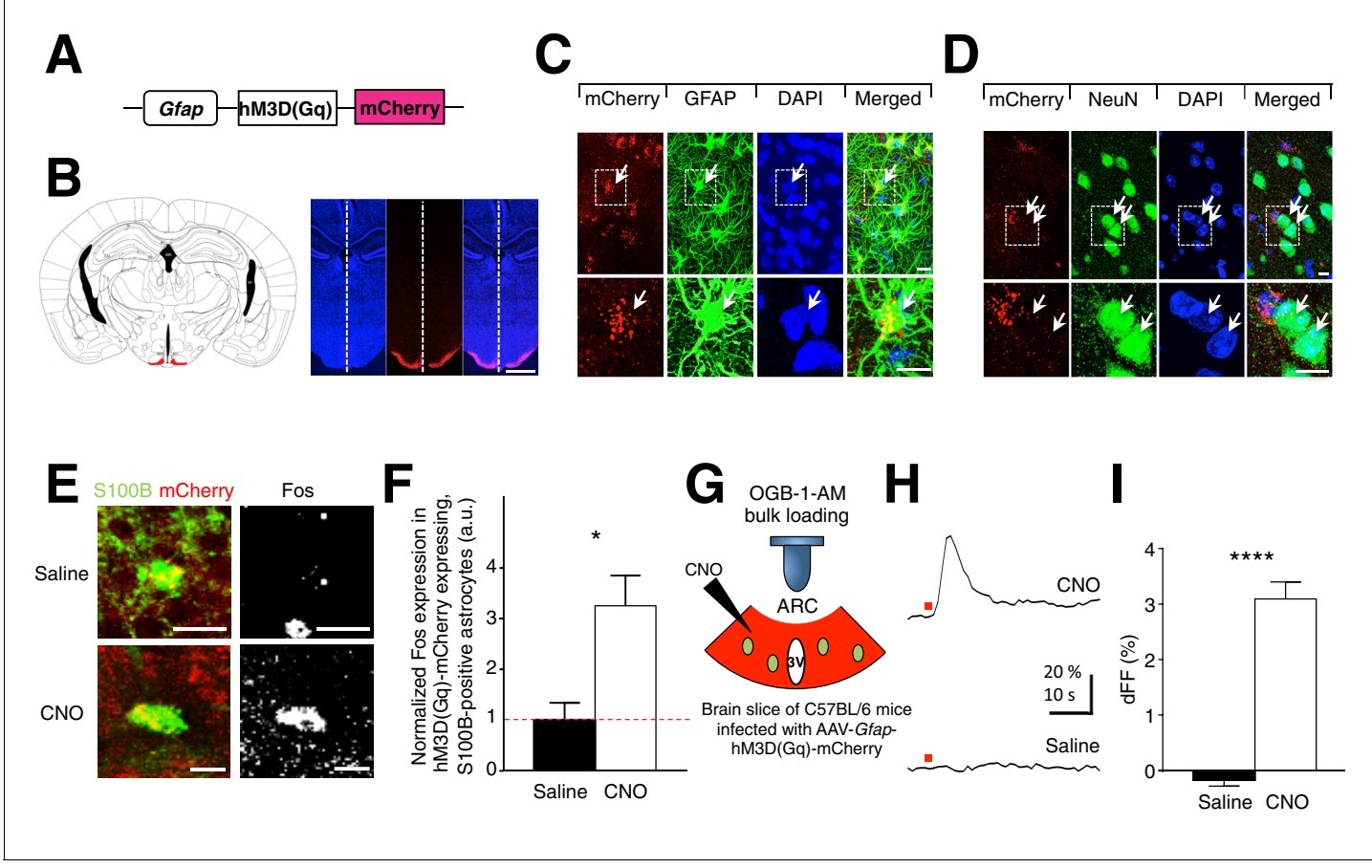

**Figure 1.** CNO-dependent activation of hM3D(Gq)-mCherry expressing ARC glia evokes increased Fos immunoreactivity and elevated intracellular $Ca^{2+}$. (A) Design of the AAV construct expressing hM3D(Gq)-mCherry under the *Gfap* promoter. (B) (Left) Schematic drawing showing the location of ARC (red) in a coronal brain slice (Right) An example DAPI-stained (blue) coronal brain slice containing hM3D(Gq)-mCherry (red) expressing glia in ARC. Scale bar, 1 mm. (C–D) DAPI, anti-GFAP and anti-NeuN immunohistochemistry showing expression of hM3D(Gq)-mCherry in glia but not in neurons. (Inset) Magnified image of a hM3D(Gq)-mCherry expressing glial cell and non hM3D(Gq)-mCherry expressing neurons. Scale bars, 10 µm. (E) *In vivo* injection of CNO but not saline induces Fos immunoreactivity in hM3D(Gq)-mCherry-expressing (red), S100B-positive (green) ARC astrocytes. Animals were perfused for Fos quantification 2 hr post injection. Scale bar, 10 µm. (F) Population mean of normalized Fos expression in astrocytes following saline or CNO injection. (G) Configuration of calcium imaging of OGB-1-AM loaded, hM3D(Gq)-mCherry expressing astrocytes in ARC coronal slices during CNO application. (H) Local application of CNO but not saline (red dot; 10 mM, 20 psi, 200 ms) evoked a robust $Ca^{2+}$ increase in ARC astrocytes. (I) The population average of mean% fluorescence change (dFF) of astrocytes when CNO or saline was applied. *p<0.05, ****p<0.0001. Error bars represent SEM. See also *Figure 1—figure supplement 1*.

The following figure supplement is available for figure 1:

**Figure supplement 1.** DREADDs are specifically expressed in glial cells. Astrocytes, but not tanycytes, are activated by DREADDs and fasting. Morphological changes in astrocytes are also induced by DREADDs and fasting.

hM3D(Gq)-mCherry was selectively expressed in glia and not in neurons. This was evidenced by the co-localization of mCherry with anti-GFAP (*Figure 1C*) and anti-S100B (*Figure 1—figure supplement 1A*: 93.9% of mCherry-expressing cells (*n* = 401/430 cells in 12 animals) were immunopositive for S100B) but not with anti-neuronal nuclei (NeuN) immunohistochemistry stains (*Figure 1D*). There was also no expression of mCherry in specific neuronal sub-types: orexigenic AgRP/NPY neurons and anorexigenic POMC neurons in *Npy*-hrGFP and *Pomc*-EGFP mice respectively (*Figure 1—figure supplement 1A*: 99.1% of GFP-expressing neurons in *Npy*-hrGFP mice (*n* = 733/744 neurons in 11 animals) and 99.3% of GFP-expressing neurons in *Pomc*-EGFP mice (*n* = 439/448 neurons in 9 animals) did not express mCherry). The hM3D(Gq)-mCherry glial population comprised both astrocytes and tanycytes that line the third ventricle (*Figure 1—figure supplement 1B*: 14.9% of the mCherry-

expressing cells co-localized with anti-vimentin, a marker for tanycytes ($n$ = 80/512 mCherry-expressing cells in 4 animals). *In vivo* injections of CNO (0.3 mg/kg) (*Koch et al., 2015*; *Krashes et al., 2011*, *2013*; *Pei et al., 2014*) induced significantly greater Fos immunoreactivity (*Palkovits et al., 2007*; *Ramírez et al., 2015*) as compared to saline injections in ARC astrocytes (*Figure 1E–F*: $n$ = 95 mCherry-expressing astrocytes in 5 animals each for saline and CNO groups, p=0.0114, unpaired *t*-test, comparing averaged responses across astrocytes in each saline- and CNO-injected animal) but not in tanycytes (*Figure 1—figure supplement 1C*). CNO, but not saline application in ARC slices (*Figure 1G*) also evoked robust $Ca^{2+}$ responses in hM3D(Gq)-mCherry-expressing astrocytes (*Figure 1H–I*: $n$ = 42 mCherry-expressing astrocytes in 3 animals each for saline and CNO groups, p=5.49E-16, unpaired *t*-test, comparing averaged responses across astrocytes in the saline- and CNO-injected animals). A greater complexity of astrocytic processes was also observed in CNO-injected animals as compared to saline-injected animals (*Figure 1—figure supplement 1D–F*).

Notably, the physiological activation of ARC during fasting also induced Fos immunoreactivity in ARC astrocytes and neurons but not in tanycytes (*Figure 1—figure supplement 1G*). A greater complexity of astrocytic processes was also observed in fasted animals as compared to fed animals (*Figure 1—figure supplement 1H–J*). These data suggest that activation of hM3D(Gq)-mCherry in astrocytes can induce both the functional and morphological changes observed during physiological activation of astrocytes after fasting. Collectively, these results confirm the specific and selective expression and activation of hM3D(Gq)-mCherry in ARC glia (mainly astrocytes), thereby allowing investigation of the functional effects of physiological activation of ARC glia.

## Acute activation of ARC glia reversibly induces increased food intake

To assess the effect of ARC glial activation on feeding, we compared the food intake of viral-injected, *ad libitum* fed mice in custom-designed cages (*Figure 2—figure supplement 1A*) on days with CNO injection to that under control conditions. The control conditions included (1) cage acclimatization without injection (baseline), (2) with saline injection before CNO injection (pre-CNO saline) and (3) after CNO injection (post-CNO saline). The comparison of food intake following saline and CNO injections allowed the same animal to be used as its own control (*Figure 2A*), since CNO, but not saline, can activate hM3D(Gq)-mCherry-expressing glia specifically (*Figure 1E–I*) (*Agulhon et al., 2013*). The CNO dose was selected to be 0.3 mg/kg (*Koch et al., 2015*; *Krashes et al., 2011*, *2013*; *Pei et al., 2014*) for all experiments, as concentration experiments in control mice with low (0.3 mg/kg) and high (5 mg/kg) (*Yang et al., 2015*) CNO concentrations revealed non-specific inhibition of food intake at high but not low CNO concentration (*Figure 2—figure supplement 1B–D*). Saline or CNO injections were performed at 09:00, 2 hr after the start of light phase when mice were in a calorically replete state. Feeding measurements were made between 09:00–17:00. Interestingly, the average total food intake during days of CNO administration was about three- to four-fold greater than that during baseline, pre-CNO saline and post-CNO saline (*Figure 2B*: p<0.0001, $n$ = 11 animals, unpaired *t*-test comparing total food intake during three days of CNO with that during three days of baseline, three days of pre-CNO saline and four days of post-CNO saline). This feeding response was reversible and reproducible across repeated CNO administration in the same animal (*Figure 2B*), suggesting that glial activation may serve as an intrinsic regulatory mechanism of food intake.

The CNO effect was also time-dependent when the population and trial-averaged food intake at specific time points between 09:00 and 17:00 during CNO and saline administration were examined. The time course revealed increased food intake within 30 min post CNO injection with peak feeding within 1 hr post injection. This effect persisted for at least 3 more hours (*Figure 2C*: $n$ = 11 animals, two-way analysis of variance (ANOVA), Drug: p<0.0001, $F_{(2,30)}$ = 19.0; Time: p<0.0001, $F_{(8,240)}$ = 9.71; Interaction between drug and time: p<0.0001, $F_{(16,240)}$ = 11.6; see also *Figure 2—figure supplement 1E*). At peak feeding, the food intake was more than 13 times greater than that during saline administration (*Figure 2C*). It is worth mentioning that CNO administration evoked increased food intake within 3 hr post CNO application in the dark phase when majority of food intake occurs (*Figure 2—figure supplement 2A–C*). The total food intake in the 8 hr of dark phase was however not significantly different between CNO and saline application (*Figure 2—figure supplement 2D*). This may be due to a ceiling effect in the drive to feed when glia may already be activated at night (see section below).

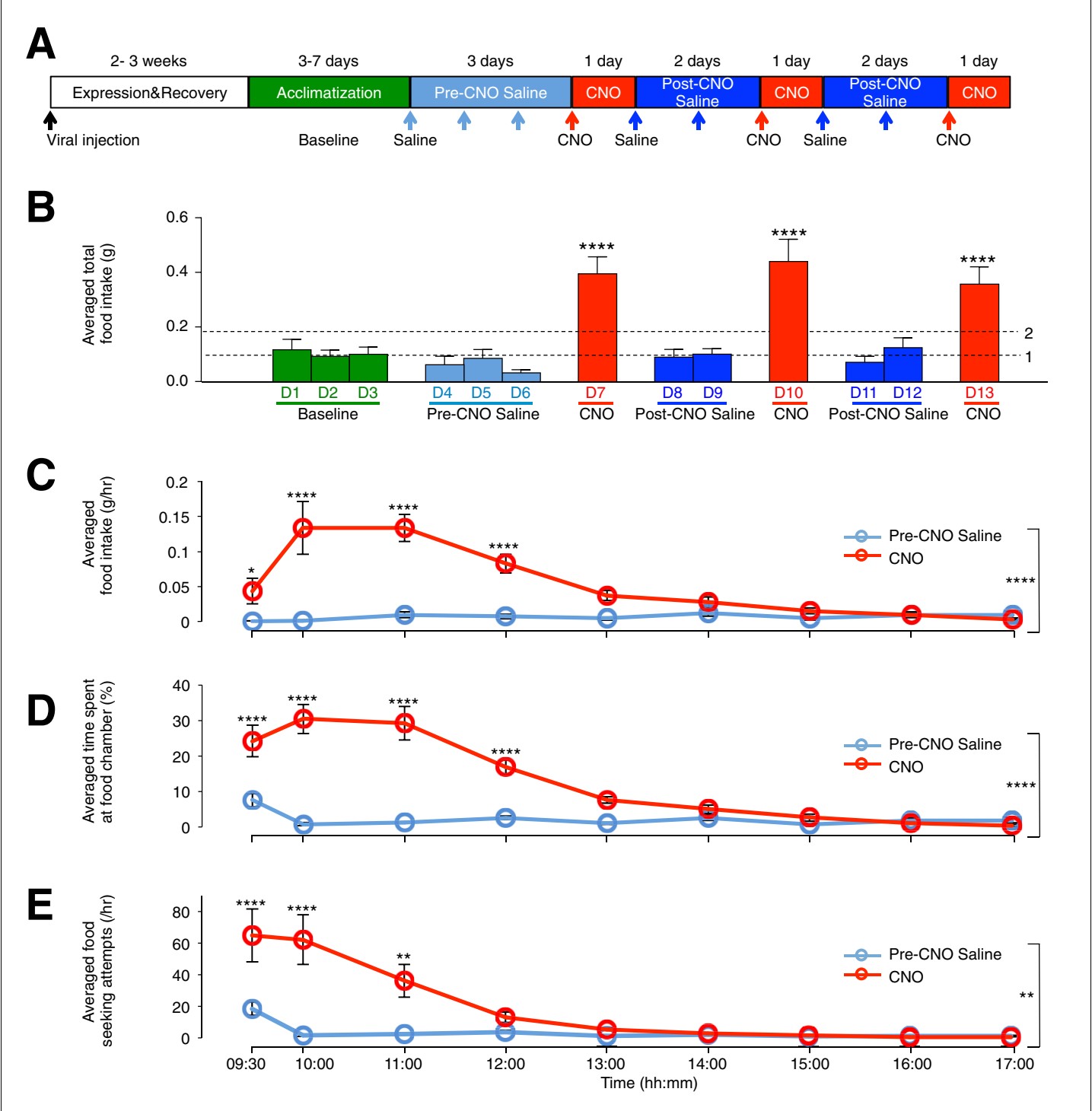

**Figure 2.** CNO-dependent activation of hM3D(Gq)-mCherry expressing ARC glia evokes increased day food intake, time spent at food chamber and food seeking attempts in C57BL/6 mice. (A) Schematic of experimental paradigm. The mice were allowed to recover and for hM3D(Gq)-mCherry to be expressed 2–3 weeks post viral injection before acclimatization in custom cages for 3–7 days (baseline) and to saline injection for three days (pre-CNO saline). CNO injections were repeated for three days, each separated by two days of saline injection (post-CNO saline) to allow the CNO effects to clear. All injections were performed at 09:00 while the food intake was measured at specific time points between 09:00–17:00. All mice were housed in custom cages between 09:00–17:00 and returned to the standard cages after 17:00 daily. (B) Total food intake between 09:00–17:00 during baseline, pre-CNO saline, CNO and post-CNO saline averaged across animals. Dotted line 1 refers to the averaged total food intake across baseline, pre- and post- CNO saline while dotted line 2 refers to two folds of this average. (C) Food intake during hourly time points from 09:00 to 17:00 (except from 09:00–09:30 and 09:30–10:00 where 30 min time points were used) during pre-CNO saline and CNO administration. (D) The percentage of time mice

*Figure 2 continued on next page*

*Figure 2 continued*

spent at food chamber relative to other cage areas during specific time points following pre-CNO saline and CNO administration. (**E**) The frequency of attempts made to access the food chamber during specific time points following pre-CNO saline and CNO administration. In *Figure 2C–E*, values between 09:00–09:30 and 09:30–10:00 were normalized to hourly values. Pre-CNO saline and CNO values were averaged across three days of repeats before computing the average across animals. Two-way ANOVA followed by Bonferroni post hoc tests was used. *p<0.05, **p<0.01, ****p<0.0001. Error bars represent SEM. See also *Figure 2—figure supplements 1–6*.

The following figure supplements are available for figure 2:

**Figure supplement 1.** CNO administered at 5 mg/kg (but not at 0.3 mg/kg) induces non-specific effects on feeding in animals without DREADD expression. CNO(0.3 mg/kg) evoked robust feeding behavior in animals with hM3D(Gq)-mCherry-expressing ARC glia.

**Figure supplement 2.** Activation of ARC glia evokes increased food intake within 3 hr after CNO application (0.3 mg/kg) but does not increase total food intake in the dark phase.

**Figure supplement 3.** Analysis of viral expression showed that mCherry is specifically expressed within the ARC, although there were a few cases where the expression spread into adjacent areas in the hypothalamus.

**Figure supplement 4.** Administration of CNO (0.3 mg/kg) in mice lacking hM3D(Gq)-mCherry in GFAP-expressing ARC glia does not evoke increased food intake, frequency of feeding attempts and duration of feeding.

**Figure supplement 5.** Administration of CNO (0.3 mg/kg) in control mice with mCherry-expressing ARC glia does not evoke increased food intake.

**Figure supplement 6.** DREADD activation of glia does not lead to changes in body weight.

We next compared the percentage of time viral-injected mice spent at the food chamber relative to other areas of the cage after CNO and saline administration (*Figure 2D*). The average percentage of time that mice spent at the food chamber following CNO administration was significantly greater than that following saline administration in the same mice (*Figure 2D*: $n = 11$ animals, ANOVA, Drug: $p<0.0001$, $F_{(1,20)} = 47.8$; Time: $p<0.0001$, $F_{(8,160)} = 26.6$; Interaction between drug and time: $p<0.0001$, $F_{(8,160)} = 22.7$, comparing the population and trial-averaged time spent at food chamber at specific time points between 09:00 and 17:00 during pre-CNO saline and CNO administration). The same was true for the frequency of food seeking attempts (*Figure 2E:* $n = 11$ animals, ANOVA, Drug: $p=0.003$, $F_{(1,20)} = 11.4$; Time: $p<0.0001$, $F_{(8,160)} = 17.2$; Interaction between drug and time: $p<0.0001$, $F_{(8,160)} = 10.1$, comparing the population and trial-averaged frequency of attempts at specific time points between 09:00 and 17:00 during pre-CNO saline and CNO administration). CNO administration also increased the frequency of shorter-duration feeding attempts and increased the maximum duration of the feeding episodes (*Figure 2—figure supplement 1F*). Collectively, these observations suggest that glial activation can trigger mice to devote greater time for feeding.

A trend of increased CNO-induced food intake was observed with increased size of hM3D(Gq)-mCherry expression in the ARC (*Figure 2—figure supplement 3*). In mice lacking hM3D(Gq)-mCherry expression in ARC due to failed viral injection attempts (identified in post hoc analysis) (*Figure 2—figure supplement 4*) as well as mice injected with the control virus AAV-*Gfap*-mCherry in ARC (*Figure 2—figure supplement 5*), CNO administration did not increase food intake. Although glial activation resulted in the pronounced feeding response, DREADD activation of glia did not lead to changes in the body weight (*Figure 2—figure supplement 6*).

Collectively, these data support an active role of ARC glia in the modulation of feeding behavior and suggest that direct glial activation is sufficient to evoke acute feeding even during the daytime when the mice are normally resting and in calorically replete state.

## Disruption of Ca²⁺ signaling pathway in ARC glia reduces food intake

The presence of activated ARC astrocytes during fasting (*Figure 1—figure supplement 1G*) suggests that glia may play a critical role in modulating feeding under physiological conditions. To address this question, we assessed the effect of reduced ARC glial activity on feeding by first injecting an AAV comprising a *Gfap* promoter-driven gene that encodes the pleckstrin homology (PH)

domain of phospholipase C (PLC)-like protein p130 (p130PH) fused with mRFP (AAV-*Gfap*-p130PH-mRFP) in ARC. Glial expression of the p130PH construct was previously shown to disrupt the $Ca^{2+}$ signaling in astrocytes *in vivo* by acting as a mobile cytosolic $IP_3$ buffer to inhibit release of $Ca^{2+}$ from internal stores (*Xie et al., 2010*). Control animals were injected with AAV-*Gfap*-mRFP without the p130PH construct (*Figure 3A*). We compared the Fos activity of mRFP-expressing, S100B-positive ARC glia in the AAV-*Gfap*-p130PH-mRFP and AAV-*Gfap*-mRFP (control) injected animals after fasting. Indeed, the Fos activity of astrocytes in the AAV-*Gfap*-p130PH-mRFP injected animals was significantly lower than that in the control animals (*Figure 3B,C*: p130PH: *n* = 847 mRFP-expressing astrocytes, Control: *n* = 508 mRFP-expressing astrocytes, 8 animals each, p=0.0264, unpaired *t*-test, comparing averaged responses across astrocytes in each *Gfap*-p130PH-mRFP and control animal). The Fos activity of tanycytes, however, was not significantly different between the experimental and control groups (*Figure 3—figure supplement 1A*). The Fos activity of neurons was significantly lower in experimental than in control animals (*Figure 3—figure supplement 1B*), suggesting possible glial-neuronal modulation (see below).

We then compared the food intake of the *Gfap*-p130PH-mRFP-injected animals and control animals during both dark and light phases. Interestingly, the population and trial-averaged food intake was significantly lower in *Gfap*-p130PH-mRFP-injected animals than control animals in the dark phase but not light phase (*Figure 4A*: *n* = 8 animals, ANOVA, Drug: p<0.0001, $F_{(1,14)}$ = 30.3; Time: p<0.0001, $F_{(19,266)}$ = 72.9; Interaction between drug and time: p=0.0131, $F_{(19,266)}$ = 1.92). This reduction of food intake during the dark phase was not compensated by an increase in food intake in the light phase as the cumulative food intake of *Gfap*-p130PH-mRFP-injected animals at the 24-hr time point (19:00) remained significantly lower than control animals (*Figure 4B*: *n* = 8 animals, ANOVA, Drug: p<0.0001, $F_{(1,14)}$ = 44.7; Time: p<0.0001, $F_{(19,266)}$ = 1091; Interaction between drug and time: p<0.0001, $F_{(19,266)}$ = 16.4). Similar to the DREADD experiments, no significant difference in the body weight was observed between *Gfap*-p130PH-mRFP-injected animals and control animals (*Figure 3—*

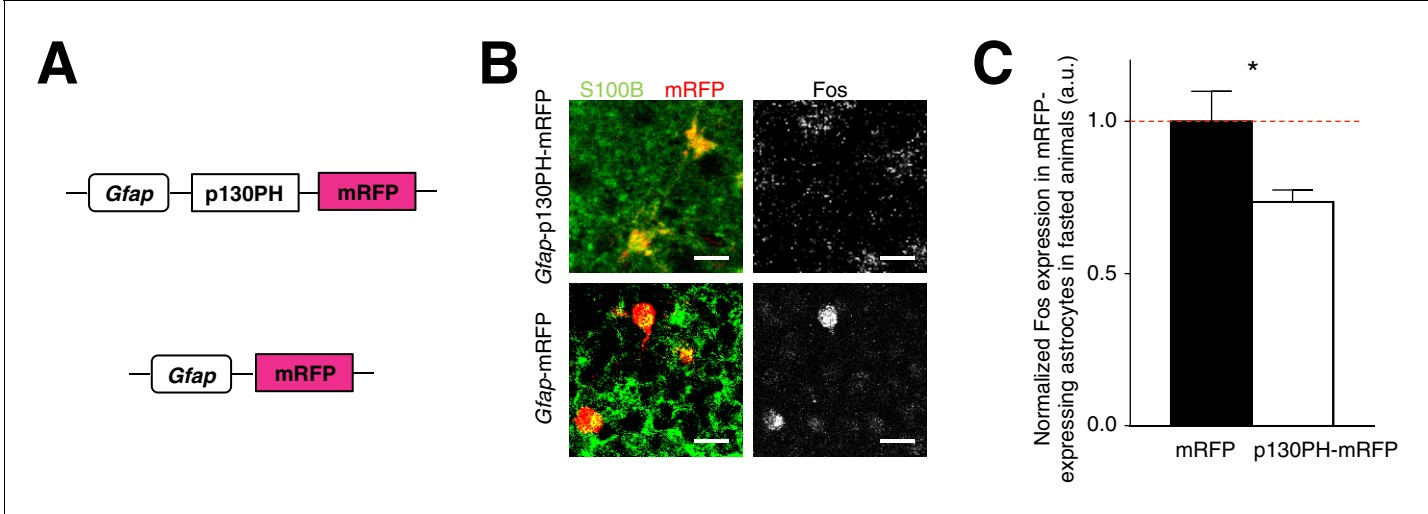

**Figure 3.** Disruption of $Ca^{2+}$ signaling in ARC glia with selective expression of p130PH-mRFP leads to decreased Fos immunoreactivity. (**A**) Design of the AAV constructs expressing *Gfap* promoter-driven p130PH-mRFP and mRFP (control) respectively. (**B**) Fos immunoreactivity in mRFP-expressing (red), S100B-positive (green) ARC astrocytes is lower in fasted animals injected with AAV-*Gfap*-p130PH-mRFP than in control animals injected with AAV-*Gfap*-mRFP. Animals were perfused for Fos quantification 16–18 hr post initiation of fasting. Scale bar, 20 μm. (**C**) Population mean of normalized Fos expression in ARC astrocytes of AAV-*Gfap*-p130PH-mRFP and AAV-*Gfap*-mRFP injected animals. *p<0.05. Error bars represent SEM. See also *Figure 3—figure supplement 1*.

The following figure supplement is available for figure 3:

**Figure supplement 1.** Disruption of $Ca^{2+}$ signaling in ARC glia with selective expression of p130PH-mRFP leads to decreased Fos immunoreactivity in neurons (but not in tanycytes) with no change in the body weight of fasted animals.

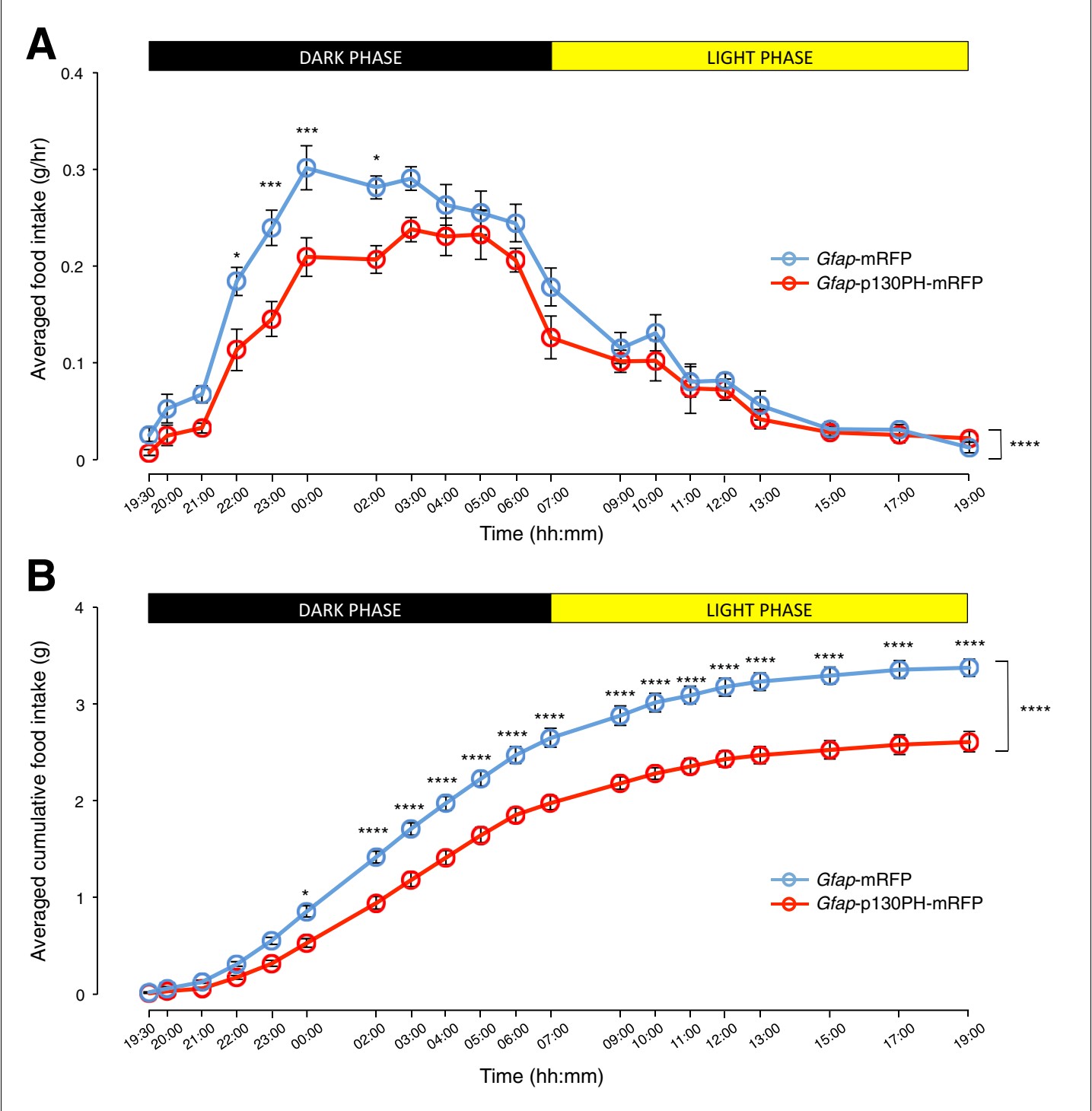

**Figure 4.** Disruption of Ca$^{2+}$ signaling in ARC glia with selective expression of p130PH-mRFP leads to decreased food intake in C57BL/6 mice during the dark phase. (**A**) Food intake during hourly time points (except from 19:00–19:30 and 19:30–20:00 where 30 min time points were used) in AAV-*Gfap*-p130PH-mRFP or AAV-*Gfap*-mRFP injected animals. Values between 19:00–19:30 and 19:30–20:00 were normalized to hourly values. (**B**) Cumulative food intake from 19:00–19:00 (24 hr, both dark and light phases) at the specific time points in AAV-*Gfap*-p130PH-mRFP or AAV-*Gfap*-mRFP injected animals. Values were averaged across three days of repeats before computing average across animals. Two-way ANOVA followed by Bonferroni post hoc tests was used. *p<0.05, ***p<0.001, ****p<0.0001. Error bars represent SEM.

*figure supplement 1C*). Collectively, these findings support an essential role for glia in the modulation of feeding during physiological conditions.

## Acute activation of glia facilitates the activity of AgRP/NPY neurons but not POMC neurons

There is emerging evidence that astrocytes are integral components of brain circuits linked to specific brain functions and are capable of modulating neuronal responses (*Eroglu and Barres, 2010*; *Halassa and Haydon, 2010*). It is thus possible that the activation of DREADD-expressing glia can consequently modulate nearby ARC neurons to contribute to the observed evoked feeding behavior.

To test if this mechanism underlies the glial activation-induced increase in feeding, we investigated the potential role of the glial modulation of AgRP/NPY and POMC neurons. We first injected the hM3D(Gq)-mCherry virus in the ARC of *Npy*-hrGFP mice and *Pomc*-EGFP mice. Next, we performed *ex vivo* whole-cell current clamp recordings from NPY neurons (GFP-positive neurons in *Npy*-hrGFP mice) and POMC neurons (GFP-positive neurons in *Pomc*-EGFP mice) in brain slices containing hM3D(Gq)-mCherry-expressing ARC glia (*Figure 5A*). Interestingly, CNO activation of the hM3D(Gq)-mCherry-expressing ARC glia depolarized and/or increased the firing rate of AgRP/NPY neurons (*Figure 5B,C*: $n$ = 15 neurons in 15 slices from 9 animals, p<0.0001, unpaired $t$-test, comparing CNO-induced responses with null responses; averaged resting membrane potential: $-52.2 \pm 2.65$ mV [*van den Top et al., 2004*]). In POMC neurons, CNO did not induce a significant response at a population level (*Figure 5B,C*: $n$ = 14 neurons in 14 slices in 6 animals, p=0.370; unpaired $t$-test, comparing CNO-induced responses with null responses; averaged resting membrane potential: -44.1 $\pm$ 2.09 mV [*Cowley et al., 2001*]). These findings were further validated by first injecting both *Npy*-hrGFP and *Pomc*-EGFP mice (with hM3D(Gq)-mCherry-expressing ARC glia) with either CNO or saline before performing Fos immunohistochemistry. Indeed, greater Fos immunoreactivity was observed after CNO injection as compared to saline injection in NPY but not POMC neurons (*Figure 5D–E,F*: $n$ = 5 *Npy*-hrGFP animals each for CNO and saline injections, p=0.0428, unpaired $t$-test, comparing CNO and saline induced responses; $n$ = 5 and n = 6 *Pomc*-EGFP animals for CNO and saline injections respectively, p=0.736, unpaired $t$-test, comparing CNO and saline induced responses). Collectively, these observations show that direct and acute activation of ARC glia facilitates the activity of AgRP/NPY but not POMC neurons.

To confirm that the CNO-evoked facilitatory responses in NPY neurons indeed have an astrocytic origin, we performed chelation of astrocytic $Ca^{2+}$ by patch-clamping electrophysiologically characterized astrocytes (*Figure 5—figure supplement 1A*) with the cell-impermeable $Ca^{2+}$ chelator 1,2-Bis(2-aminophenoxy) ethane-$N,N,N',N'$-tetraacetic acid (BAPTA). The spread of BAPTA within the local syncytium of astrocytes through gap junctions (*Jourdain et al., 2007*), visualized by including Alexa Fluor 633 (A633) in the patch pipette, was confirmed to be about 200 µm from the patched astrocyte after 20–50 min of dialysis (*Figure 5—figure supplement 1B*). Indeed, BAPTA chelation of $Ca^{2+}$ in hM3D(Gq)-mCherry-expressing ARC astrocytes blocked the CNO-evoked depolarization in NPY neurons (*Figure 5—figure supplement 1C–D*: Control: $n$ = 15 neurons in 15 slices in 9 animals, BAPTA: $n$ = 5 neurons (patched after BAPTA dialysis of astrocyte syncytium) in 5 slices in 3 animals, p=0.0050, unpaired $t$-test, comparing responses in the absence and presence of BAPTA dialysis). Together these findings demonstrate that astrocytic $Ca^{2+}$ activation contributes to NPY neuronal facilitation.

## POMC neurons receive inhibitory inputs that mask the facilitation induced by ARC glial activation

Although we have observed preferential activation of AgRP/NPY neurons over POMC neurons during ARC astrocytic activation, it is unclear how this cell-type specific modulation arises. Is this due to the intimate organization between astrocytes and specific neuronal subtypes? Or do astrocytes indiscriminately facilitate both AgRP/NPY and POMC neurons but their net responses shaped by the presence of a strong AgRP to POMC inhibition (*Atasoy et al., 2012*)? To dissect these possibilities, we first recorded spontaneous inhibitory postsynaptic currents (sIPSCs) from POMC neurons during CNO activation of hM3D(Gq)-mCherry-expressing ARC glia. Indeed, CNO evoked an increase in the frequency (but not amplitude) of sIPSCs in POMC neurons (*Figure 6A top*, 6B: $n$ = 7 neurons in 7

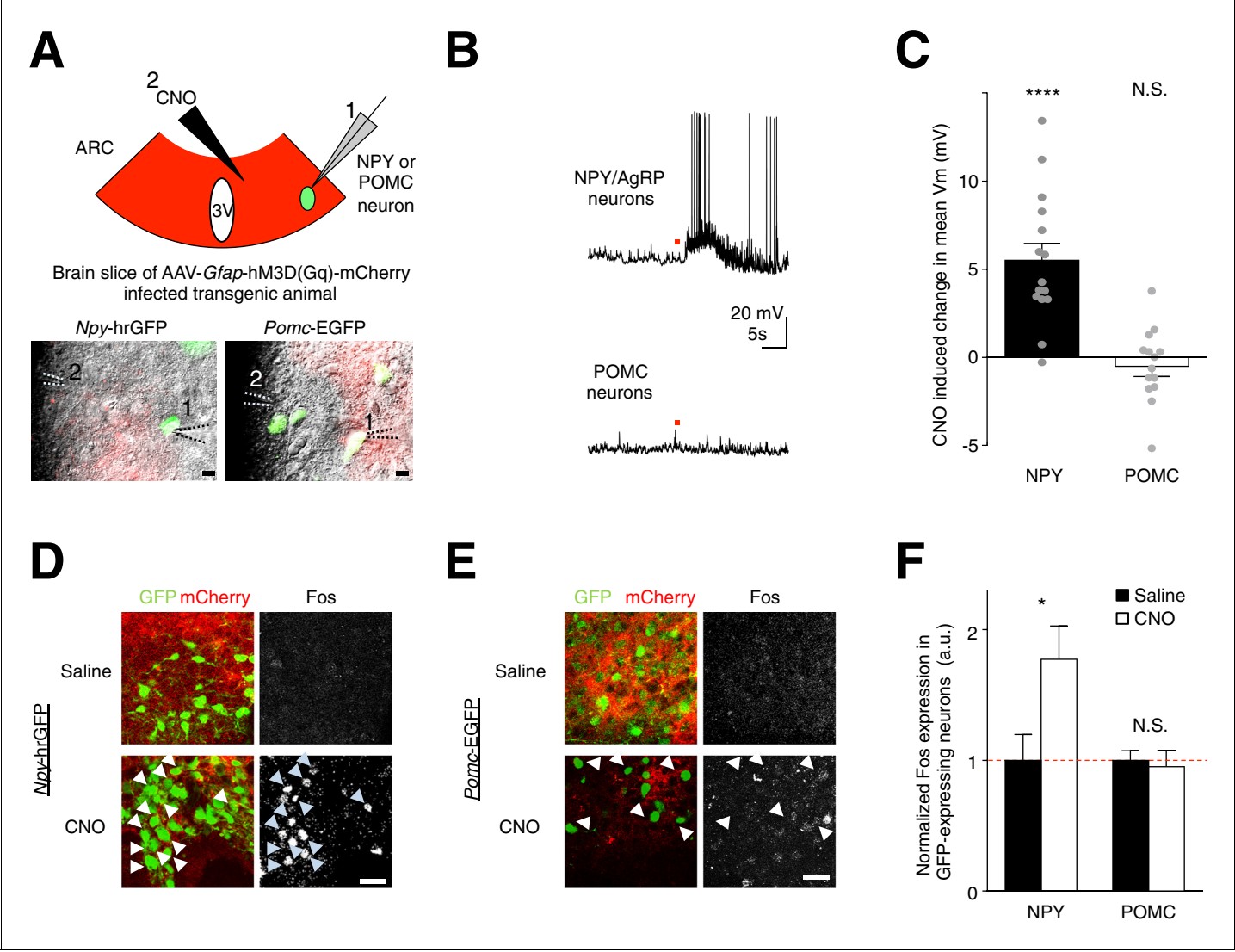

**Figure 5.** CNO activation of hM3D(Gq)-expressing glia evokes facilitatory responses in NPY but not POMC neurons. (**A**) (**Top**) Configuration of whole-cell patch-clamp recordings of GFP-labeled NPY or POMC neurons in the ARC of coronal brain slices infected with AAV-*Gfap*-hM3D(Gq)-mCherry during CNO application. (**Bottom**) Merged green fluorescence (GFP), red fluorescence (mCherry) and differential interference contrast images of (**Left**) a GFP positive NPY neuron and (**Right**) a GFP positive POMC neuron patched in viral injected *Npy*-hrGFP and *Pomc*-EGFP slices respectively. Relative positions of (1) patch pipette and (2) CNO drug pipette were as indicated. Scale bars, 10 μm. (**B**) Local CNO application (red dot; 10 mM, 20 psi, 200 ms) evoked depolarizing response in (**Top**) an NPY (**Bottom**) but not in POMC neuron. (**C**) A population average of mean membrane potential ($V_m$) of NPY or POMC neurons when CNO was applied. (**D–E**) *In vivo* injection of CNO induced greater Fos immunoreactivity in NPY neurons than saline injection. This was not observed in POMC neurons. Scale bars, 20 μm. (**F**) Population mean of normalized Fos expression in NPY and POMC neurons following saline or CNO injection. *$p < 0.05$, ****$p < 0.0001$, N.S., not significant. Error bars represent SEM. See also *Figure 5—figure supplement 1–2*

The following figure supplements are available for figure 5:

**Figure supplement 1.** CNO does not evoke any response in NPY neurons when calcium is chelated in hM3D (Gq)-mCherry-expressing ARC astrocytes.

**Figure supplement 2.** Administration of CNO (0.3 mg/kg) in Agrp-Ires-cre mice with hM4D(Gi)-mCherry-expressing AgRP neurons and hM3D(Gq)-mCherry-expressing ARC glia does not evoke any change in food intake.

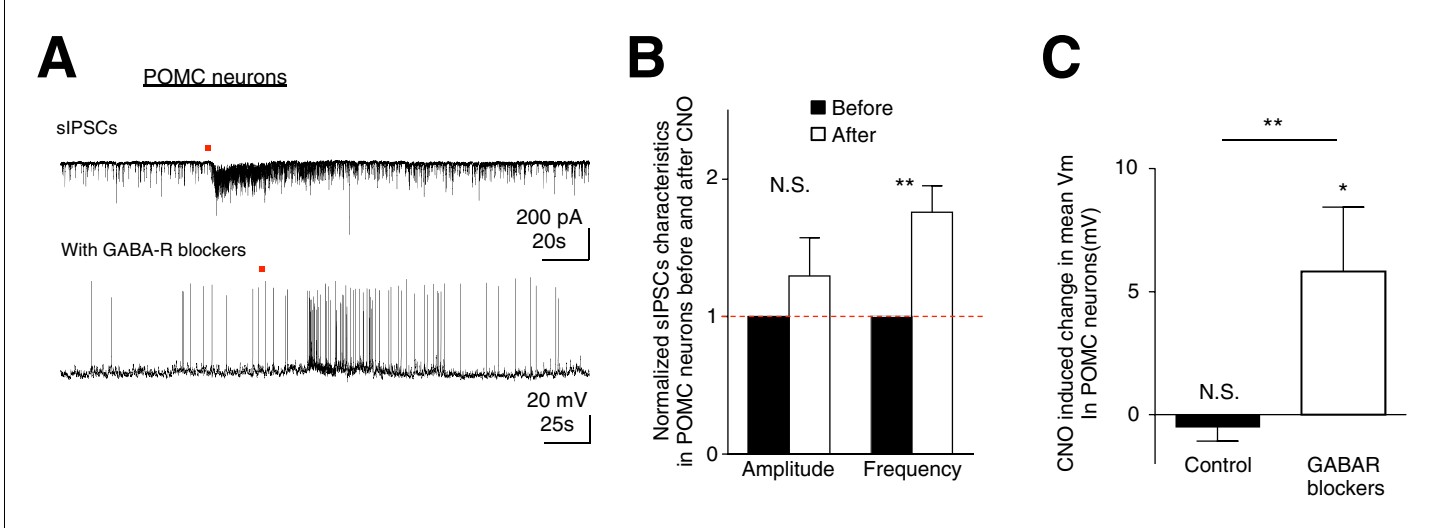

**Figure 6.** CNO activation of hM3D(Gq)-expressing glia evokes increased frequency of sIPSCs in POMC neurons. In the presence of GABAergic blockers, hM3D(Gq)-expressing glia activated depolarization in POMC neurons is revealed. (**A**) (*Top*) Local CNO application (red dot; 10 mM, 20 psi, 200 ms) evoked increased frequency of sIPSCs in POMC neurons patched in high chloride internal solution in the presence of bath application of NBQX. (**Bottom**) Local CNO application (red dot; 10 mM, 20 psi, 200 ms) evoked a facilitatory response in POMC neurons patched in the presence of picrotoxin and SCH-50911 (see Materials and methods: Slice physiology). (**B**) Population average of mean normalized amplitude and frequency of sIPSCs in POMC neurons before and after CNO application. (**C**) Population average of mean $V_m$ of POMC neurons when CNO was applied in the absence and presence of GABAergic blockers (p=0.370 and p=0.0460; unpaired *t*-test, comparing CNO-induced responses with null responses). *p<0.05, **p<0.01, N.S., not significant. Error bars represent SEM.

The following figure supplement is available for figure 6:

**Figure supplement 1.** Administration of CNO (0.3 mg/kg) in mice with hM4D(Gi)-mCherry-expressing ARC glia does not evoke any change in food intake.

slices in 3 animals, Amplitude: p=0.324; Frequency: p=0.0078, paired *t*-test, comparing normalized responses before and after CNO application). This observation reveals strong inhibition received by POMC neurons during glial activation, possibly via NPY neuronal activation. To test if facilitatory responses in POMC neurons evoked by glial activation can be unmasked when the strong inhibition they receive is blocked, we recorded the CNO-evoked responses in POMC neurons in the presence of GABA receptor (GABAR) blockers (50 μM Picrotoxin, a GABA$_A$ receptor antagonist and 20 μM SCH-50911, a GABA$_B$ receptor antagonist, see Materials and methods: Slice physiology). Interestingly, CNO evoked strong depolarization in POMC neurons when patched in the presence of GABAR blockers (*Figure 6A bottom*, *Figure 6C*: Control: n = 14 neurons in 14 slices in 6 animals, GABAR blockers: n = 7 neurons in 7 slices in 5 animals, p=0.0046, unpaired *t*-test, comparing responses between in the absence and presence of GABAR blockers). These data support our hypothesis that astrocytic activation facilitates both AgRP/NPY and POMC neurons indiscriminately. Unlike AgRP/NPY neurons, however, the presence of a strong inhibitory input, possibly due to the activation of AgRP/NPY-POMC inhibitory connection (*Atasoy et al., 2012*) during glial activation, can further shape the net responses of POMC neurons. Our data thus suggest that ARC glial activation indiscriminately facilitates both AgRP/NPY neurons and POMC neurons but the presence of an inhibitory input leads to no net response in POMC neurons, thereby providing a mechanistic understanding of the glial activation-induced increase in feeding.

## Acute activation of ARC glia while inactivating AgRP/NPY neurons does not increase food intake

To test if AgRP/NPY activation is necessary for glial activation-induced increase in feeding, we measured the food intake during simultaneous activation of ARC glia and inactivation of AgRP/NPY neurons. This was achieved by stereotaxic injection of both AAV-*Gfap*-hM3D(Gq)-mCherry and AAV-

hSyn-DIO-hM4D(Gi)-mCherry in the ARC of Agrp-Ires-cre mice that allows selective expression of hM3D(Gq)-mCherry in ARC glia and hM4D(Gi)-mCherry in AgRP/NPY neurons (*Figure 5—figure supplement 2A–B*). The food intake of these viral-injected, *ad libitum* fed mice was then compared on days with CNO and saline injections (*Figure 5—figure supplement 2C*).

The average total food intake (*Figure 5—figure supplement 2D*) as well as population and trial-averaged time course of food intake (*Figure 5—figure supplement 2E*) during days of CNO administration were not significantly different from those during saline administration. This finding that AgRP/NPY activation is required to mediate the glial activation-induced increase in feeding further establishes the crucial role of the ARC glia-AgRP/NPY circuit in the regulation of energy balance.

## Discussion

Our work shows that ARC glial activation enables increase in food intake via AgRP/NPY neurons. Interestingly, ARC glial activation facilitates both AgRP/NPY and POMC neurons. Unlike AgRP/NPY neurons, however, the responses in POMC neurons are further shaped by an inhibitory input, possibly due to activation of the AgRP/NPY neuron-POMC neuron inhibitory pathway (*Atasoy et al., 2012*). The individual POMC neuronal responses are variable (see *Figure 5c*), suggesting varying relative strengths of the astrocyte-POMC and astrocyte-AgRP/NPY-POMC pathways. These competing pathways, however balance, revealing no significant response at the population level. Our findings show that although glial cells interact with distinct neuronal subtypes non-specifically, it can confer neuronal subtype-specific modulation through direct modulation of distinct neuronal circuits. This may be a common mechanism to explain the glial-neuronal subtype specific modulation observed in other brain regions (*Perea et al., 2014*).

The lack of astrocyte-neuronal subtype wiring also suggests that glial activation may modulate other neuronal subtypes within the ARC. This thus opens the possibility of glial modulation of energy expenditure-promoting neurons in ARC e.g. GABAergic RIP-Cre neurons which do not overlap with the POMC and AgRP/NPY neuronal population (*Kong et al., 2012*) and can potentially prevent the expected body weight gains during AgRP/NPY neuronal activation (*Krashes et al., 2011*). This may explain the absence of weight changes during ARC glial modulation observed in our study (*Figure 2—figure supplement 6*, *Figure 3—figure supplement 1C*).

There are a few possible mechanisms that may underlie the non-specific, glial activation-stimulation of AgRP neurons and POMC neurons. These include: (1) astrocytic release of glutamate or D-serine gliotransmitters (*Halassa et al., 2007*; *Haydon and Carmignoto, 2006*; *Parpura and Zorec, 2010*; *Scofield et al., 2015*; *Volterra and Meldolesi, 2005*); (2) regulation of extracellular transmitters (*Pannasch et al., 2014*) or extracellular potassium (*Wang et al., 2012*) possibly through changes in the activity of transporters in the astrocyte membrane (*Bazargani and Attwell, 2016*). Further investigation is required to dissect these possibilities.

Under physiological conditions, glial activation may occur via neuronal-glial interactions (*Allen and Barres, 2009*; *Bazargani and Attwell, 2016*; *Fields and Burnstock, 2006*; *Lalo et al., 2006*), possibly evoked by active AgRP/NPY neurons in the presence of a strong drive to feed (*Chen et al., 2015*; *Mandelblat-Cerf et al., 2015*). This, together with the strong glial-neuronal interactions observed in our work (*Figure 5A–C*, *Figure 1—figure supplement 1G*, *Figure 3B–C*, *Figure 3—figure supplement 1B*) suggest modulation of feeding by bidirectional communication (*Allen and Barres, 2009*) between glia and neurons. Direct glial modulation by orexigenic molecules (*Murphy and Bloom, 2006*) such as ghrelin via astrocytic ghrelin receptors (*García-Cáceres et al., 2014*) may also be a possible mechanism to glial activation.

We manipulated glial $Ca^{2+}$ activity bi-directionally with viral-mediated Gq-DREADD technology (*Agulhon et al., 2013*; *Armbruster et al., 2007*) and viral-mediated p130PH disruption of glial $Ca^{2+}$ signaling (*Xie et al., 2015*, *2010*). Both methods allow local and specific manipulation of $Ca^{2+}$-signaling pathway of glia in ARC. The Gq-DREADD technology mimics acute activation of astrocytic GPCRs that triggers $Ca^{2+}$ activation (*Agulhon et al., 2013*, *2008*) while the p130PH technology enables disruption of the endogenous PLC/IP$_3$ glial $Ca^{2+}$ signaling (*Li et al., 2013*; *Xie et al., 2015*, *2010*) during physiological feeding states. We have concerns using the Gi-DREADD technology to manipulate glial activity. Although the Gi-DREADD technology has been shown to inhibit neuronal firing via induction of hyperpolarization and inhibition of presynaptic neurotransmitter release (*Roth, 2016*), it is unclear whether Gi-DREADD can affect glial activity or glial $Ca^{2+}$ signaling

pathways as these cells are electrically non-excitable and do not fire action potentials. In addition, existing work that activated Gi-coupled GPCR pathways in glia do not reveal any blockade of both astrocytic activity and astrocyte-neuronal modulation (*Agulhon et al., 2012*). Indeed, we did not observe any effect of glial Gi-DREADD activation on feeding (*Figure 6—figure supplement 1*).

Previous work has provided evidence that support the role for astrocytes in feeding. This includes experiments showing chronic modification of feeding by synaptic remodeling (*Horvath et al., 2010*) during conditional deletion of astrocytic leptin receptors (*Kim et al., 2014*) (see also *Jayaram et al., 2013*) as well as observation of reactive gliosis (*Horvath et al., 2010*) and activation of astrocytic inflammatory signaling pathways after feeding of high fat diet (*Buckman et al., 2015*). These studies however do not address the question of whether direct glial-activation can evoke acute changes in feeding. The behavioral changes in feeding in these studies also cannot be solely isolated to the hypothalamus due to body-wide deletion of leptin receptors in GFAP-expressing cells (expression of GFAP (*Apte et al., 1998*; *Buniatian et al., 1998*; *Nolte et al., 2001*) and leptin receptors (*Bjørbaek and Kahn, 2004*; *Gautron and Elmquist, 2011*) occur in both the central nervous system and periphery). Our work directly addresses these gaps in knowledge by demonstrating a causal relationship between arcuate glia and acute as well as chronic modulation of food intake. We also further provide mechanistic evidence to explain how glial activation can differentially modulate AgRP/NPY and POMC neurons through direct modulation of distinct neuronal pathways and demonstrate the necessity of active AgRP/NPY neurons in mediating the glial activation-induced feeding.

A previous study (*Yang et al., 2015*) investigating glial regulation of feeding by using the CNO-DREADD system reported distinct conclusions from our study. We attribute the differences to the use of different CNO doses and show that a high CNO dose (5 mg/kg) used in that study (*Yang et al., 2015*) can in fact induce non-specific inhibition in feeding (see *Figure 2—figure supplement 1B–D*), possibly due to CNO-induced ataxia (Sigma-Aldrich's toxicological studies [*Sigma-Aldrich, 2016*]).

Our work, together with previous findings, collectively demonstrates a critical role of glia in the regulation of feeding and further establishes ARC as an important site of glial-mediated regulation. These findings also imply a possible causal link between increased ARC glial $Ca^{2+}$ during astrogliosis (*Kanemaru et al., 2013*) and hyperphagia during ARC inflammation (*Dorfman and Thaler, 2015*). Future work in this area may reveal glia as a possible target of therapeutic intervention.

## Materials and methods

### Mice

All mice were housed at room temperature with a 12-hr light and 12-hr dark cycle (lights on: 07:00; lights off: 19:00). Mice were housed with cage mates except after surgery, during food behavior, Fos, glial morphological quantification and acute slice experiments where they were housed individually. For food behavior experiments, C57BL/6 (Taconic, Germantown, NY) and *Agrp*-Ires-cre (*Agrp*$^{tm1(cre)Lowl}$/J; JAX 012899; Bradford B. Lowell, Harvard) adult mice of both gender, between 7–14 weeks old were used. For Fos experiments, acute slice whole cell patch-clamp, calcium imaging and immunohistochemistry experiments, C57BL/6, *Npy*-hrGFP (*van den Pol et al., 2009*) (B6.FVB-Tg [*Npy*-hrGFP]1Lowl/J; JAX 006417; Bradford B. Lowell, Harvard), *Pomc*-EGFP (*Cowley et al., 2001*) (C57BL/6J-Tg(*Pomc*-EGFP)1Low/J; JAX 009593; Malcolm J. Low, University of Michigan Medical School) mice of both gender, between 7 weeks–8 months old were used. All mice were housed in standard cages except during behavior experiments where custom-designed cages with no bedding (*Figure 2—figure supplement 1A*) were used. Standard mouse chow in pellet form (Prolab RMH 3000, 5P00*; Gross energy: 4.19 kcal/g, Metabolizable energy: 3.18 kcal/g, Physiological fuel value: 3.46 kcal/g; Protein: 25.999%, Fat: 14.276%, Carbohydrates: 59.725%) and water were provided *ad libitum.* All experiments were performed under protocols (0513-044-16) approved by the Animal Care and Use Committee at MIT and conformed to NIH guidelines.

### Viral construct and injection

The AAV-*Gfap*-hM3D(Gq)-mCherry virus (Construct: Bryan Roth, UNC Chapel Hill, Addgene plasmid # 50478; AAV Serotype 2/8, UNC Vector Core), AAV-*Gfap*104-mCherry virus (Construct: Ed Boyden, MIT, AAV Serotype 2/8, UNC Vector Core), AAV-*Gfap*-p130PH-mRFP and AAV-*Gfap*-mRFP viruses

(Shinghua Ding (*Xie et al., 2010*), Serotype 2/5), AAV-hSyn-DIO-hM4D(Gi)-mCherry virus (Construct: Bryan Roth, UNC Chapel Hill, Addgene plasmid # 44362; AAV Serotype 2/8, UNC Vector Core) and AAV-*Gfap*-hM4D(Gi)-mCherry virus (Construct: Bryan Roth, UNC Chapel Hill, Addgene plasmid # 50479; AAV Serotype 2/5, UNC Vector Core) were injected and specifically expressed in ARC glia. Mice were initially anesthetized with 4% isoflurane in oxygen and maintained on 1.5–2% isoflurane on a stereotaxic apparatus. The skull was exposed and a small hole was drilled above each side of ARC. Bilateral injection was performed with a glass micropipette (20–30 µm diameter) filled with 200 nl of virus at the following coordinates: bregma: AP:−1.40 mm, DV:−5.80 mm, L: ±0.30 mm. The injection speed was controlled at 100 nl/min with a micromanipulator (Quintessential Stereotaxic Injector, Stoelting). Mice were injected intraperitoneally with meloxicam (1 mg/kg) for postoperative care. Experiments were performed 2–3 weeks post-injection to allow for recovery and viral expression.

## Food intake and feeding behavior studies

*Ad libitum* fed mice were transferred from their standard cages to the open-top, custom-designed cages (*Figure 2—figure supplement 1A*) daily during the duration of the study. Each cage contained a food chamber, a drinking spout (water bottle external to the cage), mouse igloo and sterile gauze to substitute the removed bedding. The mouse igloo was positioned opposite to the food chamber in each cage and served to be an alternative location for rest and play. The weight of the food chamber was taken at specific time points using a weighing balance (CAS MWP-300N, 300 g, 0.01 g precision). In some experiments, measurements were taken after intraperitoneal injections of saline or clozapine-N-oxide (CNO, C0832, Sigma-Aldrich, St. Louis, MO) at 09:00 (*Figure 2B-E*, *Figure 2—figure supplement 1E–F*, *Figure 2—figure supplements 4–5*, *Figure 5—figure supplement 2*), 19:00 (*Figure 2—figure supplement 1B–D*, *Figure 6—figure supplement 1*) or 22:00 (*Figure 2—figure supplement 2*). The feeding behavior was continuously recorded during the experiments with web cameras (Logitech, Newark, CA) positioned above the cages and the open source iSpy webcam software (iSpyConnect, Perth, Australia). Mice were transferred back to the standard cages after these experiments. CNO was administered at 0.30 mg/kg of body weight (see *Figure 2—figure supplement 1B–D*) while saline was given at the same volume as control.

## Post behavior experiment localization of viral expression and immunohistochemistry

At the conclusion of the behavior study, all mice were perfused and the fixed brains were sectioned and imaged to confirm the stereotaxic accuracy of the viral injection site. Some of the sectioned slices were immunostained to verify the expression specificity of hM3D(Gq)-mCherry in glia. **Perfusion**: Mice were anesthetized with 4% isoflurane and perfused transcardially with 0.1 M PBS followed by chilled 4% paraformaldehyde in 0.1 M PBS. The brains were then postfixed in 4% paraformaldehyde in 0.1 M PBS (<4°C) overnight. Fixed brains were sectioned into 50–70 µm coronal slices (containing ARC) with a vibratome. **Localization of viral expression**: The fixed slices were mounted on a glass slide with the Vectashield Hardset mounting media (Vector Labs, Burlingame, CA). The slides were imaged using a confocal microscope (Zeiss LSM 5 Pascal Exciter,Carl Zeiss, Oberkochen, Germany) and the images were analyzed for localization of mCherry (*Figures 1–2*, *Figure 5—figure supplement 2*) or mRFP (*Figures 3–4*) expression in ARC. **Anti-GFAP/Anti-S100B and anti-NeuN immunohistochemistry**: The fixed slices were blocked in 10% normal goat serum with 0.5% triton in PBS (1 hr, room temperature) before being stained with mouse anti-GFAP (1:400,MAB360; Millipore, Billerica, MA) or mouse anti-S100B (1:600, S2532; Sigma) and rabbit anti-NeuN (1:250, ABN78; Millipore) in blocking buffer overnight (at 4°C). This was followed by a 3-hr incubation in Alexa Fluor 488 goat anti-mouse (1:200, Invitrogen, Carlsbad, CA, A11029) and Alexa Fluor 488 goat anti-rabbit (1:250, Invitrogen, A11034) in 1% normal goat serum and 0.3% triton, before being mounted on a glass slide with mounting media. Confocal imaging was performed and the images were analyzed for presence of co-localization of mCherry/mRFP and GFAP /S100B (*Figure 1C,E*, *Figure 3B*, *Figure 1—figure supplement 1A*, *Figure 5—figure supplement 2B*), absence/presence of co-localization of mCherry and NeuN (*Figure 1D*, *Figure 5—figure supplement 2B*).

## Anti-Fos immunohistochemistry and morphological analysis of glia

Mice were perfused 2 hr post CNO/saline injection or 16–18 hr post initiation of fasting in fasted animals. The brains were post-fixed in paraformaldehyde over 2 nights before being sectioned into 100 µm coronal slices. The fixed slices were first incubated in 1.2% Triton-X-100 for 15 min before being blocked in 5% normal goat serum with 2% BSA and 0.2% Triton-X-100 in PBS (1 hr, room temperature). For anti-Fos immunohistochemistry, the slices were then incubated in blocking buffer containing rabbit anti-Fos (1:1000, PC38; Calbiochem, San Diego, CA) and in some experiments with mouse anti-S100B (1:600, S2532; Sigma) overnight at 4°C. For morphological analysis of glia, slices were incubated in blocking buffer containing rabbit anti-GFAP (1:400, G9269, Sigma) overnight at 4°C. This was followed by a 2-hr incubation in blocking buffer containing Alexa Fluor 647 goat anti-rabbit (1: 1000, Invitrogen, A21245) and Alexa Fluor 488 goat anti-mouse (in some anti-Fos immunohistochemistry experiments, 1:1000, Invitrogen, A11001) as well as 0.3 µM DAPI in PBS before being mounted on a glass slide with mounting media. Confocal imaging was performed with similar optical parameters for all slices imaged to ensure fair comparisons.

For anti-Fos immunohistochemistry experiment, MetaMorph Basic Offline (v. 7.7.0.0, Molecular Devices, Sunnyvale, California) was used for image quantification. Individual astrocytic glial cells were identified by the S100B and in some cases with mCherry/mRFP labeling (*Figure 1E-F*, *Figure 3B–C*, *Figure 1—figure supplement 1G*) while NPY/AgRP and POMC neurons were identified by their GFP expression (*Figure 5D-F*). In some cases, putative neurons were identified by DAPI and their lack of S100B and/or mRFP labeling (*Figure 1—figure supplement 1G* and *Figure 3—figure supplement 1B*). Tanycytic glial cells were identified by their location along the third ventricle next to the ARC (*Figure 1—figure supplement 1C*, *Figure 1—figure supplement 1G*, *Figure 3—figure supplement 1A*) and mCherry/mRFP labeling (*Figure 1—figure supplement 1C*, *Figure 3—figure supplement 1A*). These cells were then circled manually before the mean Fos signal for each cell was computed. Normalized Fos values were computed by normalizing raw Fos signal by the averaged Fos signal across control cells.

For morphological analysis of glia (*Reeves et al., 2011*), ARC glia with clear soma and processes were identified by GFAP labeling (*Figure 1—figure supplement 1D*, *Figure 1—figure supplement 1H*) and in some cases with mCherry expression (*Figure 1—figure supplement 1D*). Image stacks in 1 µm planes were then collected under 60x oil objective with 2x digital zoom. The glial cells were 10–90 µm from the surface. Image stacks were traced and 3D-reconstructed using the Neurolucida software (MBF Bioscience, Williston, VT). The coordinate files generated by the reconstruction were further analyzed by Sholl analysis using Neuroexplorer software (MBF Bioscience, Williston, VT). Virtual circles at increasing radii of 3.9 µm increments were drawn from the soma of each glia. The total length of GFAP-labeled processes passing through each circle and the number of intersections of processes with each circle were quantified (*Figure 1—figure supplement 1E–1F*, *Figure 1—figure supplement 1I–J*)

## Anti-vimentin immunohistochemistry

100 µm coronal fixed slices, prepared as described above, were stained with mouse monoclonal anti-vimentin (2 µg/ml, DSHB Hybridoma Product 40E-C, deposited to DSHB (Iowa City, Iowa) by Alvarez-Buylla, Arturo) followed by Alexa Fluor 488 goat anti-mouse (1:1000, Invitrogen, A11001).

## Food intake and behavior analysis

The amount of food intake during a specific time segment for each mouse was computed by subtracting the weight of the food chamber at the beginning of the time point by that at the end of the time point. The time segment for total food intake was from 09:00–17:00 (*Figure 2B*, *Figure 2—figure supplement 4C*) or 09:00–13:00 (*Figure 2—figure supplement 5C*) or 09:00–15:00 (*Figure 5—figure supplement 2D*) or 19:00–23:00 (*Figure 6—figure supplement 1C*) or 22:00–06:00 (*Figure 2—figure supplement 2D*). The total food intake data was averaged across animals (*Figure 2B*, *Figure 2—figure supplement 2D*, *Figure 2—figure supplement 4C*, *Figure 2—figure supplement 5C*, *Figure 5—figure supplement 2D*, *Figure 6—figure supplement 1C*) as well as averaged across trials and animals (*Figure 2C*, *Figure 4*, *Figure 2—figure supplement 1C–E*, *Figure 2—figure supplement 2B–C*, *Figure 2—figure supplement 4D–E*, *Figure 2—figure supplement 5D*, *Figure 5—figure supplement 2E*, *Figure 6—figure supplement 1D–E*). Processing of the videos was done

using in-house code written in MATLAB (Natick, MA). The average percentage of time spent at the food chamber was computed as (total time spent at the chamber in the time segment)/(total duration in the time segment)*100% for each animal before taking the average across trials and animals (*Figure 2D*, *Figure 2—figure supplement 4F*). The average frequency of food-seeking attempts (/hr) was computed by counting the number of times each mouse approach (and stay) at the food chamber in the time segment before normalizing the count by the duration of the time segment considered before taking the average across trials and animals (*Figure 2E*, *Figure 2—figure supplement 4G*). The frequency-histogram was computed by binning the food-seeking attempts for all animals by the duration spent on the food chamber (*Figure 2—figure supplement 1F*, *Figure 2—figure supplement 4H*).

## Slice physiology

Coronal brain slices of ARC (300 µm) were prepared from AAV-*Gfap*-hM3D(Gq)-mCherry injected *Npy*-GFP and *Pomc*-EGFP mice with a vibratome (Leica VT 1200S, Leica, Wetzlar, Germany). Mice were anesthetized with Avertin solution (20 mg/ml, 0.5 mg/g body weight) and transcardially perfused with 15–20 ml of ice-cold carbongenated (comprising 95% $O_2$/5% $CO_2$, pH 7.33–7.38) cutting solution containing (in mM): 194 sucrose, 30 NaCl, 4.5 KCl, 1.2 NaH2PO4, 0.2 CaCl2, 8 MgCl2, 26 NaHCO3, and 10 D-(+)-glucose (360 mOsm). The slices were incubated in artificial cerebral spinal fluid (aCSF) at 32°C for 10 min followed by in fresh aCSF at room temperature for at least 1 hr. These slices were then transferred to a slice recording chamber for electrophysiology. The aCSF contained (in mM): 119 NaCl, 2.3 KCl, 1.0 NaH2PO4, 26.2 NaHCO3, 11 Glucose, 1.3 MgSO4, 2.5 CaCl2 (pH 7.4, 295–305 mOsm).

Whole-cell patch clamp recordings were performed with IR-DIC. All recordings were conducted at room temperature. The electrophysiological current-clamp recordings were filtered at 10 kHz and sampled at 10 kHz at gain = 1 while that for voltage-clamp recordings were filtered at 2 kHz and sampled at 10 kHz at gain = 5. The intracellular pipette solution for patching neurons in *Figure 5A–C*, *Figure 5—figure supplement 1C–D*, *Figure 6A* bottom, *Figure 6C* contained (in mM): 131 KGluconate, 17.5 KCl, 9 NaCl, 1 MgCl2.6H2O, 10 HEPES, 1.1 EGTA, 2 MgATP, 0.2 Na2GTP (pH 7.3, 300 mOsm). The intracellular pipette solution for patching astrocytes in *Figure 5—figure supplement 1A* contained (in mM): 50 KGluconate, 8 NaCl, 1 MgCl2, 10 HEPES, 13 K2SO4, 2 MgATP, 0.4 NaGTP, 40 BAPTA tetrapotassium salt (sc-202076, Santa Cruz Biotech, Texas, USA) (pH 7.3, 300 mOsm). Alexa Fluor 633 Hydrazide (A633, A30634, ThermoFisher Scientific, MA, USA) was also included to allow the observation of the spread of BAPTA in the astrocyte syncytium. The A633 was imaged with an orange-red excitation filter/far red emission filter. The intracellular pipette solution for patching neurons in *Figure 6A* top and *Figure 6B* contained (in mM): 103 CsCl, 12 Cs-methanesulfate, 5 TEA-Cl, 4 NaCl, 10 HEPES, 0.5 EGTA, 4 MgATP, 0.3 Na2GTP, 10 Phosphocreatine, 5 Lidocaine N-ethyl chloride (pH 7.4, 300 mOsm). Glass pipettes (3–5 MΩ, KG33, King Precision) were pulled with a Sutter P-97 puller (Sutter instruments). CNO (10 mM) was applied by pressure injection with a picospritzer (20 psi, 200 ms). In *Figure 6A* top, *Figure 6B*, 2,3-Dioxo-6-nitro-1,2,3,4-tetrahydrobenzo[*f*]quinoxaline-7-sulfonamide (10 µM NBQX, ab120045, Abcam, MA, USA) was bath-applied. In *Figure 6A* bottom, *Figure 6C*, picrotoxin (50 µM PTX, 1128, Tocris, Bristol, UK) was included in the intracellular pipette solution while SCH 50911 (20 µM, 9084, Tocris, Bristol, UK) was bath applied. In a select number of neurons, both PTX and SCH 50911 were included in the intracellular pipette solution and the data are also included in the population data in *Figure 6C*. GFP labeled NPY or POMC ARC neurons were visualized with an Olympus BX61WI microscope (Olympus, Tokyo, Japan) coupled with a 40x water immersion lens (Olympus). Recordings were performed with a multiclamp 700B amplifier and digidata 1440A data acquisition system, with pClamp software in the current-clamp mode. Analysis was performed with the Clampfit 10.2.0.12 software (Molecular Devices). The input resistances of the AgRP and POMC neurons patched in *Figures 5–6*, *Figure 5—figure supplement 1C–D* were comparable and agreed with previous literature (*Fu and van den Pol, 2010*). The CNO-induced change in mean $V_m$ in *Figures 5C, 6C* and *Figure 5—figure supplement 1D* was defined as (Averaged membrane potential) $_{after\ drug}$ - (Averaged membrane potential) $_{before\ drug}$ where (Averaged membrane potential) was computed as the averaged membrane potential over 10 s. In *Figure 6B*, the amplitude and frequency of each event in the 120 s segment before and after CNO application was quantified before the population average was taken over all events in the respective segments. The population averaged amplitude or frequency of each

segment was then normalized by the population averaged amplitude or frequency computed across events within the 120 s segment before CNO application.

## Slice calcium imaging

Coronal brain slices of ARC (300 µm) were prepared from AAV-*Gfap*-hM3D(Gq)-mCherry injected C57BL/6 mice as described above. The slices were placed on a porous membrane in a 3.5 mm dish bathed in carbogenated aCSF. Oregon Green 488 BAPTA-1 AM (OGB) dye solution was prepared by adding 1 µl Pluronic F-127 (20% Solution in DMSO, P3000MP, ThermoFisher Scientific, MA, USA) and 49 µl DMSO to 50 µg of OGB (O6807, ThermoFisher Scientific). Bulk loading of the dye solution was performed by adding 3 µl of this dye solution over the ARC. After incubation at room temperature in the dark for 40 min, the slices were transferred to fresh aCSF for an additional 30 min before the experiment.

hM3D(Gq)-mCherry glia in ARC were visualized with an Olympus BX61WI upright microscope under a 40x/0.80 W Olympus LUMPlanFL N water immersion lens and an Olympus CellSens dimension (v 1.6)-driven Hamamatsu ORCA-R2 digital CCD camera (C10600). Fluorescence was continuously excited using a Lumen 200/220 (Prior Scientific, MA, USA). OGB1-AM and mCherry were imaged with a blue excitation filter/green emission filter and a green excitation filter/red emission filter, respectively. Images were continuously acquired at 1.25 Hz, using 1x1 binning. Images were saved by the CellSens software as tagged image fileformat files and analyzed with both ImageJ (NIH) and custom-written MATLAB scripts. Individual cells were circled manually based on the mCherry expression. The mean fluorescence for each cell was computed from frame to frame, giving a time-varying intensity signal for each cell. After correction for lamp flicker noise and dye photobleaching, dFF was calculated by subtracting the local corresponding baseline (F) from the peak of the response to get dF before taking the ratio by F. Increases in dFF in ROIs indicated an increase in $Ca^{2+}$ concentration.

## Statistics

Analysis was performed using GraphPad Prism (La Jolla, CA). Two-way ANOVA followed by Bonferroni post hoc tests and standard two-tailed paired or unpaired *t*-test were used as indicated in text. A p value of less than 0.05 was considered significant in these studies. Error bars indicate S.E.M. Blind experiments were performed for the Fos and slice physiology studies. Blind experiments were not performed for the behavior studies but the same criteria were applied to all allocated groups for comparisons. No randomization was performed for the study. No statistical methods were used to predetermine sample sizes, but our sample sizes are similar to those reported to previous publications (*Cowley et al., 2001*; *Krashes et al., 2011*; *Wu et al., 2014*).

## Acknowledgements

We thank Dr. Shinghua Ding (U Missouri) for his kind gift of the AAV-*Gfap*-p130PH-mRFP and AAV-*Gfap*-mRFP viruses. Xiangyu Zhang, Bailey Clear, Triana Dalia, Chuong Le and Michaela Ennis (MIT) for technical assistance. Dr. Hongyu Li (A*STAR), Dr. Jeremy Petravicz, Dr. Sami El-Boustani, Xian Gao (MIT), Dr. Ken McCarthy (UNC) and Dr. Bryan Roth (UNC) for providing technical advice. Dongqing Wang, Frances Corniel, Travis Emery, Liadan Gunter (MIT), Shermaine Thein, Joy Lim, Jenny See and Meng Chu Cher (A*STAR) for administrative assistance. This work was supported by the Poitras Center for Affective Disorders Research at MIT (**GF**), intramural funding from the A*STAR Biomedical Research Council (Singapore) (**WH**) as well as grants from the US National Institutes of Health (R01EY007023, R01EY018648, U01NS090473), National Science Foundation (EF1451125) and the Simons Foundation (**MS**). **BB** is supported by postdoc fellowships from Simons Center for the Social Brain at MIT and the Autism Science Foundation.

# Additional information

## Funding

| Funder | Grant reference number | Author |
|---|---|---|
| Simons Center for the Social Brain at MIT | Postdoc fellowship | Boaz Barak |
| Autism Science Foundation | Postdoc fellowship | Boaz Barak |
| National Institutes of Health | R01EY007023 | Mriganka Sur |
| National Science Foundation | EF1451125 | Mriganka Sur |
| Simons Foundation | | Mriganka Sur |
| National Institutes of Health | R01EY018648 | Mriganka Sur |
| National Institutes of Health | U01NS090473 | Mriganka Sur |
| Poitras Center for Affective Disorders Research at MIT | | Guoping Feng |
| Agency for Science, Technology and Research | | Weiping Han |

The funders had no role in study design, data collection and interpretation, or the decision to submit the work for publication.

## Author contributions

NC, Conceptualizes project, designed experiments, analyzed data and wrote analysis code, performed experiments to acquire data, wrote original draft of manuscript, reviewed and edited manuscript; HS, Designed experiments, analyzed data and wrote analysis code, performed experiments to acquire data, reviewed and edited manuscript; JK, Designed experiments, interpreted data, performed experiments to acquire data, reviewed and edited manuscript; ZF, BB, Designed experiments, analyzed data, performed experiments to acquire data, reviewed and edited manuscript; MS, Reviewing and editing manuscript, Acquired funding for project; GF, Supervision of project, Design of experiments, Reviewing and editing manuscript, Acquired funding for project; WH, Supervision of project, Reviewing and editing manuscript, Acquired funding for project

## Author ORCIDs

Naiyan Chen, http://orcid.org/0000-0002-4432-1805
Guoping Feng, http://orcid.org/0000-0002-8021-277X

## Ethics

Animal experimentation: All experiments were performed under protocols (0513-044-16) approved by the Animal Care and Use Committee at MIT and conformed to NIH guidelines. All surgical procedures were performed under anesthesia, and every effort was made to minimize suffering.

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
