## [Decision Letter]

Thank you for submitting your article "Direct modulation of GFAP-expressing glia in the arcuate nucleus bi-directionally regulates feeding" for consideration by *eLife*. Your article has been favorably evaluated by a Senior Editor and three reviewers, one of whom, Richard D Palmiter (Reviewer #1), is a member of our Board of Reviewing Editors. The following individual involved in review of your submission has agreed to reveal their identity: Michael Krashes (Reviewer #2).

The reviewers have discussed the reviews with one another and the Reviewing Editor has drafted this decision to help you prepare a revised submission.

This paper demonstrates quite convincingly that activation of arcuate astrocytes using DREADD technology can stimulate daytime feeding, while inhibition of arcuate astrocytes inhibits nighttime feeding. Furthermore, the authors provide evidence that this phenomenon is likely mediated by astrocyte activation of AGRP neurons. This is the first paper to show that astrocytes are important players in regulation of food consumption, and may explain (in part) why inflammation caused by consumption of a high-fat diet promotes weight gain if the inflammatory signals were to chronically activate astrocytes.

Below is a list of recommendations (summarized from the three reviews) that the authors should address in their revised manuscript and response.

1) The arcuate is a fairly large structure and it can be difficult to manipulate; hence, the extent of viral transduction of glia in the arcuate should be assessed. Ideally, the extent of transduction for each mouse could be correlated with the effect on appetite.

2) Glia change shape with activation. Did that occur with hM3Di activation? Are astrocytes activated to the same extent by fasting as observed with hM3Dq and CNO?

3) The loss of function studies are critical for knowing whether glia are important players in regulation of food intake. There is concern (especially in view of a previous paper by Yang et al) regarding the method used to study the effect of glia inactivation. Inclusion of studies using an inhibitory DREADD, hM4Di, at the same low dose of CNO used for the activation studies would be appropriate; alternatively explain why they were not chosen. An advantage of chronic inhibition of glia with the p130PH-mRFP method is that the long-term effect on body weight could be assessed? Those data should be included. Does chronic activation of glia affect body weight?

4) The authors provide Fos and EPhys evidence that AgRP neurons are preferentially activated in response to glial activation. Is the AgRP activation responsible for the increase in food consumption? Activating the glia while inactivating AgRP neurons would provide a definitive answer to this question. Alternatively, the authors should indicate that AgRP neuron activation is just one plausible mechanism.

5) The potential mechanism by which glial activation stimulates AgRP neurons should be discussed in more depth.

More specific comments from the reviewers that add depth to the requests above are indicated below. These comments do not require individual responses.

1) The authors state that they are selectively manipulating GFAP-expressing glia of the ARC, however this is loosely defined. The ARC is a fairly long structure spanning almost 1 mm both rostral to caudal and medial to lateral. It would be insightful if the authors made some sort of schematic to display the expression of the hM3D(Gq) virus. I imagine glia are present throughout the ARC and importantly throughout the entire hypothalamus. How certain are the authors that DREADD expression was specific for the ARC? Additional images would help convince the reader that they are selectively manipulating their intended targets. This becomes particularly critical when comparing the results here to the previous findings of Yang et al., 2015 whereby GFAP- hM3D(Gq) was expressed throughout the medialbasal hypothalamus.

2) It should be noted that the cumulative food intake reported from 9:30-17:00 is extremely low (on average of 0.1 grams; Figure 2). Although the light cycle is normally a time for calorically replete mice to rest, most reports observe feeding behavior during this time of about 10-20% of an animal's daily food intake. An explanation for this discrepancy could be the custom-designed cages versus home cage which could induce additional stressors on the animals even following the acclimation period. This very low baseline makes the CNO versus saline GFAP- hM3D(Gq) effects appear disproportionately large. Again, this could be due to the specific behavioral setup but perhaps home cage assessment of food intake may strengthen this claim.

3) It is unclear why the authors chose to use p130PH-mRFP as a method to disrupt ARC glia signaling as opposed to hM4D(Gi), the latter of which is reversible and far more acute. This tool has been used previously. Also, further characterization of the Gfap-p130PH-mRFP mice should be reported. For example, it is quite remarkable that these mice show such a sharp reduction in cumulative food intake; does this result in a leaner phenotype over an extended period of time? How does this compare to other manipulations that result in a drastic decrease in feeding behavior?

4) A proposed mechanism of AgRP/NPY, and for that matter POMC, neurons are described here. It may be worth discussing a possible glutamate-release mechanism accounting for this activation as previously described in GFAP- hM3D(Gq)-expressing glia in Scofield et al., 2015. It would also be worth discussing how a proposed glutamate release into extracellular space could be specific for certain cell-types.

5) When differentiating between the present study and that of Yang et al., 2015, the authors propose that the high CNO dose (5 mg/kg) can induce non-specific inhibition of feeding, which they show here (although baseline seems to be higher here in saline injected mice). However, although this would explain the reduction of feeding observed in the Yang et al., 2015 paper using GFAP- hM3D(Gq), it contradicts the GFAP-hM4D(Gi) elevated ghrelin-evoked feeding and blunted leptin-induced anorexia (both net gains of food intake) observed previously. Again, I think the major reason for these differences seen between these two studies is the expression of the GFAP-DREADDs, which should be more focused here.

6) This could also at least start to explain the electrophysiological differences between these two studies in which AgRP/NPY neurons are activated (here) or inhibited (Yang et al., 2015). The authors show a sample trace of CNO-induced increased frequency of AgRP/NPY neurons but should quantify this.

7) It is important to show the regulation of activities differentially in NPY/AGRP and POMC neurons by glia in the ARC and the physiological outcomes of this pathway in whole animals. However, it is not clear whether this pathway really participates in the development of obesity (body weight gain) or eating disorders (loss of body weight) in animals, which lessens the impact of this study.

8) It was not clear how glial cells activated NPY/AgRP cells and how glial activation was modulated by cues encoding the energy status of animals in this study. Without the understanding of these questions in this manuscript, the study is mostly descriptive, and potentially, reports on artifact.

9) To establish the effect of glial activation of NPY/AgRP neurons on the regulation of feeding, the authors should also demonstrate that silencing NPY/AgRP cells with an inhibitory DREADD receptor is able to eliminate the effect of glial activation with the stimulatory DREADD receptor expressed in glia on feeding.

10) To verify the results obtained in this report with a stimulatory DREADD receptor, the authors should use an inhibitory DREADD receptor to show opposite effects in mice.

11) It is known that in different energetic states (HFD, fasting) astrocytes change their morphology. In the present paper the authors investigated the role of astrocyte activation in AgRP and POMC neurons, but they obviated the effects that astrocyte activation has on the astrocytes themselves. Morphology studies on astrocytes should be carried (at least length and number of ramifications).

12) The authors compared acute activation with a fasting state. First, they don´t provide astrocyte activation after fasting. This must be measured. Then, they link the glia-neuron interaction to the calcium signaling in astrocytes. When calcium signaling in the astrocytes of the Arc is disrupted, the acute activation doesn´t have any feeding effect. If the authors said the activation has the same effects as fasting, they need to provide that fasting modulates calcium signaling in the astrocytes of the ARC. They also need to show that acute activation increases calcium signaling.

13) In order to show that astrocyte activation affects AgRP and POMC, they checked the activation measuring FOS and performing e-phys studies. In AgRP, they found increased firing, and in POMC no changes in firing, but differences in inhibitory events (maybe coming from the AgRP/NPY cells). They need to provide data of excitatory and inhibitory events onto NPY cells after CNO administration.

14) They conclude that direct modulation of astrocytes regulates feeding, but they only describe effects when they "activate" astrocytes. What is it happening when astrocytes are inactivated? Can the acute inactivation abolish feeding response to fasting? Can the acute or systemic inactivation of astrocytes impair the morphological changes triggered by fasting?

---

## [Author Response]

Below is a list of recommendations (summarized from the three reviews) that the authors should address in their revised manuscript and response.

*1) The arcuate is a fairly large structure and it can be difficult to manipulate; hence, the extent of viral transduction of glia in the arcuate should be assessed. Ideally, the extent of transduction for each mouse could be correlated with the effect on appetite.*

We have quantified the extent of viral transduction of glia in the arcuate and confirmed that the CNO-induced feeding increased with increased size of viral transduction (Figure 2—figure supplement 3).

*2) Glia change shape with activation. Did that occur with hM3Di activation? Are astrocytes activated to the same extent by fasting as observed with hM3Dq and CNO?*

We have observed a greater complexity of astrocytic processes in hM3D(Gq)-mCherry expressing arcuate glia of CNO-injected animals than in control animals injected with saline (Figure 1—figure supplement 1).

We have also observed a greater complexity of astrocytic processes in arcuate glia of fasted animals than in control fed animals (Figure 1—figure supplement 1).

Both the morphology studies (Figure 1—figure supplement 1 and Figure 1—figure supplement 1) and FOS immunoreactivity study (Figure 1 and Figure 1—figure supplement 1) suggest that activation of hM3D(Gq)-mCherry in astrocytes can induce both functional and morphological changes observed during physiological activation of astrocytes after fasting.

*3) The loss of function studies are critical for knowing whether glia are important players in regulation of food intake. There is concern (especially in view of a previous paper by Yang et al) regarding the method used to study the effect of glia inactivation. Inclusion of studies using an inhibitory DREADD, hM4Di, at the same low dose of CNO used for the activation studies would be appropriate; alternatively explain why they were not chosen.*

We performed the suggested experiment and did not observe any significant difference in the dark phase feeding after saline and CNO (0.3 mg/kg) injection in animals with hM4D (Gi)-expressing arcuate glia (Figure 6—figure supplement 1).

We have concerns with the hM4D(Gi) method because it is not known whether activating hM4D(Gi) affects astrocytic activity. In neurons, hM4D(Gi) inhibits firing via induction of hyperpolarization by Gβ/γ-mediated activation of G-protein inwardly rectifying potassium channels and inhibition of presynaptic neurotransmitters release (Roth, Neuron, 2016). The mechanisms of hM4D(Gi) activation in glia or astrocytes however remain largely unknown, as these cells are electrically non-excitable and do not fire action potentials. In addition, there is no literature that has demonstrated the effects of hM4Di activation on inactivating glial calcium pathway. Existing work that activated Gi-coupled GPCR pathways in glia also do not reveal any blockade of both astrocytic activity and astrocyte-neuronal modulation (Agulhon et al., 2012).

The p130PH-mRFP approach was chosen as it has been previously demonstrated to disrupt the PLC/IP_3_ calcium signaling in astrocytes in vivo (Xie et al., Neuroscience, 2010). In addition, we have also observed that the Fos activity is lower in p130PH-mRFP expressing glia than in control glia after fasting (Figure 3).

We have improved our discussion to explain the choice of method for glia ‘inactivation’: “We manipulated glial Ca^2+^ bi-directionally […] did not observe any effect of glial Gi-DREADD activation on feeding”.

*An advantage of chronic inhibition of glia with the p130PH-mRFP method is that the long-term effect on body weight could be assessed? Those data should be included. Does chronic activation of glia affect body weight?*

We observed no significant difference in the body weight of AAV-*Gfap*-p130PH-mRFP- and AAV-*Gfap*-mRFP-injected mice up to 11 weeks post injection (Figure 3—figure supplement 1).

Glial activation may modulate other neuronal subtypes within the ARC, which in turn trigger diverse physiological functions. Besides its effect on feeding, ARC glia may also regulate energy expenditure. Regulation of energy expenditure via glia may be mediated through GABAergic RIP-Cre neurons that also exist in the ARC and are well known as regulators of energy balance (D. Kong et al., Cell, 2012). The simultaneous regulation of feeding and energy expenditure by glia may account for the lack of change in body weight in the Gq-DREADD experiments (Figure 2—figure supplement 6) and p130PH experiments (Figure 3—figure supplement 1).

We have provided a brief discussion: “The lack of specific astrocyte-neuronal subtype wiring […] glial modulation observed in our study.”

*4) The authors provide Fos and EPhys evidence that AgRP neurons are preferentially activated in response to glial activation. Is the AgRP activation responsible for the increase in food consumption? Activating the glia while inactivating AgRP neurons would provide a definitive answer to this question. Alternatively, the authors should indicate that AgRP neuron activation is just one plausible mechanism.*

We performed this experiment and observed that astrocytic activation by CNO (as compared to control when saline was injected) did not evoke any change in food intake when AgRP/NPY neurons were inactivated in Agrp-Ires-cre mice with hM4D(Gi)-expressing AgRP neurons and hM3D(Gq)-mCherry expressing arcuate glia (Figure 5—figure supplement 2). This supports the hypothesis that AgRP neuronal activation is necessary for glial activation-induced increase in feeding.

*5) The potential mechanism by which glial activation stimulates AgRP neurons should be discussed in more depth.*

We have included a Discussion paragraph to address this: “There are a few possible mechanisms […] Future investigation is required to dissect these possibilities.”

*More specific comments from the reviewers that add depth to the requests above are indicated below. These comments do not require individual responses.*

We thank the reviewers for their specific comments. We have made the suggested language changes.